# IO-LVM: Inverse optimization latent variable models with applications to inferring and explaining paths

## Abstract

Learning representations from solutions of constrained optimization problems (COPs) with unknown cost functions is challenging, as models like (Variational) Autoencoders struggle to capture constraints to decode structured outputs. We propose an inverse optimization latent variable model (IO-LVM) that constructs a latent space of COP costs based on observed decisions, enabling the inference of feasible and meaningful solutions by reconstructing them with a COP solver. To achieve this, we leverage estimated gradients of a Fenchel-Young loss through a non-differentiable deterministic solver while shaping the embedding space. In contrast to established Inverse Optimization or Inverse Reinforcement Learning methods, which typically identify a single or context-conditioned cost function, we exploit the learned representation to capture underlying COP cost structures and identify solutions likely originating from different agents, each using distinct or slightly different cost functions when making decisions. Using both synthetic and actual ship routing data, we validate our approach through experiments on path planning problems using the Dijkstra algorithm, demonstrating the interpretability of the latent space and its effectiveness in path inference and path distribution reconstruction.

## 1 Introduction

When learning latent generative representations, it is often necessary for inferred samples to satisfy specific constraints, such as forming paths in a graph between designated start and target nodes. This requirement introduces the challenge of ensuring that the model learns the solutions of a Constrained Optimization Problem (COP). The difficulty intensifies when the feasible set of solutions is discrete, as the gradients of these solutions with respect to the model parameters are zero almost everywhere and therefore non-informative (Abbas & Swoboda, 2021).

Several previous works have focused on recovering the underlying cost of the COP that best explains the observed decisions. These are gradient-based methods that primarily address the non-informative gradient problem by either smoothing solver operations (Lahoud et al., 2024), interpolating COP solutions (Pogančić et al., 2020b), or perturbing the COP cost (Berthet et al., 2020). In the context of path planning, Inverse Reinforcement Learning (IRL) seeks to infer transition costs based on observed behavior, often by making assumptions about the probability distribution of the solution space (Ziebart et al., 2008b). Despite their contributions, a common limitation of all these methods is their inability to directly learn simultaneously from multiple agents performing different decisions. In these works, there is either an assumption of a single underlying cost or an assumption on the probability class of the COP solution.

In this paper, we introduce IO-LVM, a novel approach for learning latent representations of COP costs that can recover observed COP solutions, specifically for paths in graphs formulated as linear objective constrained problems. Our approach does not assume a single underlying COP cost, allowing it to learn effectively even when multiple agents are involved in the observed paths. The method uses amortized inference in conjunction with a black-box solver to map these costs into a meaningful and interpretable low-dimensional latent space. To address the gradient challenge, we adopt a technique similar to that of Berthet et al. (2020), perturbing the input of the black-box solver

and employing the Fenchel-Young loss (Blondel et al., 2020) to estimate the gradients of the COP solutions.

IO-LVM not only reconstructs path distributions and predicts paths for new start and target node pairs but also addresses the interpretability challenge by encoding paths into a low-dimensional latent space. In this space, similar costs are positioned close to each other, offering a more intuitive and interpretable representation of the path-planning process. This low-dimensional latent space enables new possibilities for path analysis, such as clustering latent vectors into meaningful groups, denoising paths by finding a small number of paths that covers the observed paths, or generating similar paths based a sample in the observed data. Additionally, IO-LVM allows for predicting how different agents might navigate between unseen source and target nodes, providing a flexible and robust framework for path inference in complex environments.

Traditional Variational Autoencoders (VAEs) (Kingma, 2013), although capable of encoding high-dimensional data into low-dimensional spaces, often fail to produce structured outputs, which is crucial in path planning, for instance. The decoder of a standard VAE may generate outputs that do not correspond to feasible COP solutions. Specifically, in path inference, when outputs are modeled as edge usage, the combination of edges may not form a valid path between the designated start and end nodes. Conversely, if outputs are modeled as paths themselves, the combinatorial nature of the problem results in an overwhelmingly large number of possible paths, making it impractical to account for them all. By incorporating techniques from structured prediction and amortized inference, IO-LVM ensures feasible reconstruction while preserving the interpretability characteristics inherent to VAEs.

## 1.1 OUR CONTRIBUTIONS

- We introduce IO-LVM, a method that combines variational approximation techniques with COP solver gradient estimation to learn latent representations for the underlying costs of COPs based on observed decisions, with a specific focus on paths in graphs.

- IO-LVM naturally constructs a disentangled, and sometimes multimodal, latent space, allowing for the reconstruction of observed path distributions without making assumptions about inferred paths. Notably, the ability to recover distinct (e.g., multimodal) representations for the underlying costs enables the modeling of different agents making decisions.

- We demonstrate the versatility of IO-LVM using both synthetic and real-world ship path datasets, highlighting its potential for path analysis tasks such as naturally clustering paths into meaningful groups, denoising observed paths, and predicting paths for unseen start and target nodes.

## 1.2 RELATED WORK

To address the aforementioned gradient challenge, several works have focused on differentiating through convex solvers (Amos & Kolter, 2017; Agrawal et al., 2019), enabling the construction of end-to-end learning frameworks that learn from decisions formulated as solutions to linear or quadratic programs (Donti et al., 2017; Wilder et al., 2019). However, these methods are mainly limited to continuous COP formulations and are difficult to extend to combinatorial problems such as route problems in graphs.

In addition to convex solvers, efforts to differentiate through dynamic programming algorithms have also been explored. For example, Mensch & Blondel (2018) proposed a method that specifically addresses the dynamic nature of certain COPs. Specifically for path inference, Lahoud et al. (2024) proposed differentiating through the Floyd-Warshall algorithm to learn from observed paths in graphs. However, their approach struggles with scalability as graph size increases due to the inherent complexity of the classical version of the algorithm.

Learning representations from the solutions of COPs can also be viewed as an instance of Inverse Optimization (Aswani et al., 2018; Tan et al., 2019; 2020), where the representations correspond to the cost parameters that led to the observed solutions. In various applications, such as Inverse Path Planning (Wulfmeier et al., 2017; Lahoud et al., 2024), these observed decisions are often assumed to be generated by some optimization process. However, these methods typically assume the existence

of a single underlying cost function, which may not capture the diversity of agent behaviors present in real-world scenarios.

Yet within the realm of path inference, IRL approaches (Ng et al., 2000; Ziebart et al., 2008a;b; Nguyen et al., 2015) modeled transition costs by assuming a linear mapping, learning these costs from observed paths that reflect agents' decisions. Deep IRL methods (Finn et al., 2016; Wulfmeier et al., 2017; Fernando et al., 2020) extended this framework to accommodate more complex cost functions. Nevertheless, these methods heavily rely on gradient estimation based on state visitation frequencies and do not scale well with increasing graph size, limiting their use in large-scale path-planning tasks.

Other methods, such as those proposed by Pogančić et al. (2020a) and Berthet et al. (2020), treated the COP solution as a black box and estimate gradients with respect to its inputs. Similar to our approach, on of the ideas of Berthet et al. (2020) is to utilize a Fenchel-Young loss to match inferred and observed paths within a smooth and convex space.

However, all the aforementioned methods either focus on learning a single cost function or condition this cost on a given context. They do not involve modeling a latent space that would enable to extract underlying characteristics of the structured data. IO-LVM addresses this gap by learning a latent representation of the COP with linear costs that can encode the variability in observed paths, even when multiple agents with different behaviors are involved.

Although autoencoders (Hinton & Salakhutdinov, 2006) and Variational Autoencoders (VAEs) (Kingma, 2013) have been successful in this area of learning latent representation to facilitate feature extraction and clustering, they typically struggle to decode structured outputs, which is essential for path inference tasks. A work with similar motivation to ours is that of Bentley et al. (2022), which combines VAEs with genetic algorithms. However, their method lacks a guarantee of optimality for COPs. In contrast, IO-LVM leverages gradient estimation through a specialized solver, ensuring optimality and feasibility, resulting in a more robust end-to-end learning framework.

## 2 PRELIMINARIES

In this section, we introduce the foundational concepts and techniques upon which IO-LVM is built. We begin by discussing the Evidence Lower Bound (ELBO) in latent variable models, followed by an overview of Fenchel-Young losses.

### 2.1 EVIDENCE LOWER BOUND (ELBO)

The objective in latent variable models is to perform approximate Bayesian inference, which involves estimating the posterior distribution $P(\boldsymbol{z} \mid \boldsymbol{x})$ to identify the latent variables $\boldsymbol{z}$ that best explain the observed data $\boldsymbol{x}$. However, directly computing this posterior is generally intractable. To address this, a variational distribution $q_\phi(\boldsymbol{z} \mid \boldsymbol{x})$ is introduced to approximate the true posterior. Since maximizing the exact log-likelihood of the data given the latent variables is not feasible, a lower bound, known as the Evidence Lower Bound (ELBO), on the data log-likelihood is optimized instead (Kingma, 2013; Rezende et al., 2014). The ELBO makes a trade-off between accurately reconstructing the input data (the expected log-likelihood) using a model $p_\theta(\boldsymbol{x} \mid \boldsymbol{z})$ and adhering to the prior distribution $P(\boldsymbol{z})$ for the latent variables. This trade-off is achieved through the Kullback-Leibler (KL) divergence between the variational distribution $q_\phi(\boldsymbol{z} \mid \boldsymbol{x})$ and the prior $P(\boldsymbol{z})$. Thus, the resulting loss function is the negative of ELBO:

$$l(\theta, \phi) = -\mathbb{E}_{q_\phi(\boldsymbol{z}|\boldsymbol{x})} \left[ \log p_\theta(\boldsymbol{x} \mid \boldsymbol{z}) \right] + D_{\mathrm{KL}} \left( q_\phi(\boldsymbol{z} \mid \boldsymbol{x}) \, \| \, P(\boldsymbol{z}) \right). \qquad (1)$$

### 2.2 FENCHEL-YOUNG LOSSES

Fenchel-Young losses are a class of loss functions that generalize many commonly used losses in machine learning and structured prediction (Blondel et al., 2020; Bao & Sugiyama, 2021) and are derived from the Fenchel conjugate in convex analysis (Boyd & Vandenberghe, 2004). Given an input $\boldsymbol{x}$, a score vector $\boldsymbol{y}$, and a problem formulated as $\omega(\boldsymbol{y}) \in \arg\min_{\boldsymbol{x} \in \mathcal{C}} \langle \boldsymbol{y}, \boldsymbol{x} \rangle$, the Fenchel-Young loss is defined as $l_{\mathrm{FY}}(\boldsymbol{y}, \boldsymbol{x}) = f(\boldsymbol{y}, \boldsymbol{x}) - f(\boldsymbol{y}, \hat{\boldsymbol{x}}_\Omega)$, where $\hat{\boldsymbol{x}}_\Omega$ is the regularized solution obtained from $\omega$ given the score vector $\boldsymbol{y}$, i.e., $\hat{\boldsymbol{x}}_\Omega := \omega_\Omega(\boldsymbol{y})$. The function $f(\boldsymbol{y}, \boldsymbol{x})$ represents a

scoring function that measures the value of $\boldsymbol{x}$ under the influence of $\boldsymbol{y}$. The loss compares this score to that of the regularized output $\hat{\boldsymbol{x}}_\Omega$, encouraging the solver to produce an output that aligns to the input $\boldsymbol{x}$.

One variant of the Fenchel-Young loss is obtained by transforming the optimization process $\omega(\boldsymbol{y})$ into a stochastic process by adding noise (perturbation) $\epsilon$ to the input. This introduces randomness, smoothing the objective function landscape. The perturbed Fenchel-Young loss can then be expressed as

$$l_{\text{FY}}^\epsilon(\boldsymbol{y}, \boldsymbol{x}) = f(\boldsymbol{y}, \boldsymbol{x}) - f(\boldsymbol{y}, \hat{\boldsymbol{x}}_\epsilon), \tag{2}$$

where $\hat{\boldsymbol{x}}_\epsilon := \omega(\boldsymbol{y} + \epsilon)$, and $\epsilon$ is typically drawn from a distribution that induces smoothness, such as a Gaussian. By choosing $f$ to be the original linear cost function, i.e., $f(\boldsymbol{y}, \boldsymbol{x}) = \langle \boldsymbol{y}, \boldsymbol{x} \rangle$, the gradient of equation 2 with respect to $\boldsymbol{y}$ elements becomes $\nabla l_{\text{FY}}^\epsilon(\boldsymbol{y}, \boldsymbol{x}) = \boldsymbol{x} - \hat{\boldsymbol{x}}_\epsilon$. The loss is minimized if and only if $\boldsymbol{x} = \hat{\boldsymbol{x}}_\epsilon$. For a more detailed discussion, refer to Berthet et al. (2020).

## 3 METHODS

In this section, we present IO-LVM in detail in Subsection 3.1, followed by its specific application to path planning, where the observed decisions are explicitly defined as paths in graphs, in Subsection 3.2.

### 3.1 METHOD DESCRIPTION: IO-LVM

Let $\mathcal{D} = \{\boldsymbol{x}_i\}_{i=1}^N$ be a dataset of $N$ samples, where each $\boldsymbol{x}_i \in \mathcal{X}$ and $\mathcal{X}$ is a constrained space. We interpret $\boldsymbol{x}_i$ as an optimal solution of a COP. Our main goal is to obtain a meaningful low-dimensional representation of COP costs to reconstruct COP solutions. Specifically, we aim to estimate the posterior distribution $P(\boldsymbol{z} \mid \boldsymbol{x})$, where $\boldsymbol{z} \in \mathcal{Z} \subset \mathbb{R}^k$ is a latent vector in a space of dimension $k$. Similar to VAEs, we use a nonlinear transformation $q_\phi$ to map samples $\boldsymbol{x}_i$ to the latent space $\mathcal{Z}$, and then reconstruct it back to the constrained space to ensure consistency with the original COP solution.

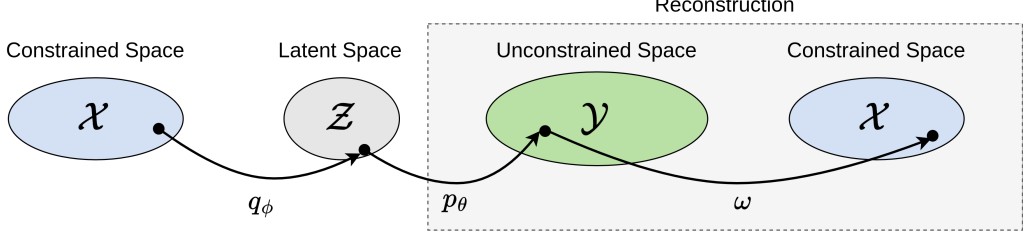

Figure 1: Proposed latent space model with a constrained reconstruction. The structured data is first mapped from $\mathcal{X}$ to a latent space $\mathcal{Z}$. The reconstruction is divided into two parts: a mapping from the latent space $\mathcal{Z}$ to an unconstrained space $\mathcal{Y}$, and another mapping from $\mathcal{Y}$ to the constrained space $\mathcal{X}$.

However, as discussed in previous sections, reconstruction in this context is non-trivial due to the constraints inherent to the COP. For example, the reconstructed output must respect the problem's structure, such as forming a valid path between specific nodes in a graph. To achieve this, we propose a reconstruction process that composes an unconstrained nonlinear transformation $p_\theta$ from the latent space $\mathcal{Z}$ to an unconstrained $\mathcal{Y}$ with a COP solver $\omega$ projecting from $\mathcal{Y}$ back to the constrained space $\mathcal{X}$. This sequence of transformations is depicted in Figure 1 and leads us to rewrite the reconstruction loss (first term) of Equation 1 as $\mathbb{E}_{q_\phi(\boldsymbol{z}|\boldsymbol{x})}\left[\mathbb{E}_{p_\theta(\boldsymbol{y}|\boldsymbol{z})}\left[d(\boldsymbol{x}, \omega(\boldsymbol{y}))\right]\right]$, where $d$ is a distance measure between the observed data in $\mathcal{D}$ and the constrained reconstructed vector in $\mathcal{X}$.

A natural choice for $d$ is the Fenchel-Young loss induced by perturbations in the input space of the COP, as described in Subsection 2.2, maintaining the reconstruction loss differentiable due to its gradient estimation. When this is done, a potential imbalance between the reconstruction loss and the regularization term can lead to one term dominating the optimization process, potentially resulting in an undesired latent representation. We address this issue by introducing a scaling factor

$\beta$ to the KL divergence, allowing us to balance the trade-off between regularizing the latent space and achieving high-quality reconstruction. This approach generalizes the concept of the $\beta$-VAE (Higgins et al., 2017; Burgess et al., 2018) to a constrained framework, where adjusting the weight of the KL term enables control over disentanglement and regularization, thus promoting a more balanced and flexible optimization. Consequently, our variational inference loss function is defined as:

$$l(\theta, \phi) = \mathbb{E}_{q_\phi(\boldsymbol{z}|\boldsymbol{x})} \left[ \mathbb{E}_{p_\theta(\boldsymbol{y}|\boldsymbol{z})} \left[ l_{\text{FY}}(\boldsymbol{x}, \omega(\boldsymbol{y})) \right] \right] + \beta D_{\text{KL}} \left( q_\phi(\boldsymbol{z}|\boldsymbol{x}) \| P(z) \right). \tag{3}$$

To learn the parameters $\theta$ and $\phi$, we minimize the empirical risk $\frac{1}{N} \sum_{i=1}^{N} l(\theta, \phi; \boldsymbol{x}_i)$ over the dataset $\mathcal{D}$. Empirically, we demonstrate that in our case, the introduction of $\beta$ also mitigates the issue of posterior collapse, which is often encountered in VAE models with powerful decoders (Van Den Oord et al., 2017). In Section 4.4, we show that the scaling factor $\beta$ tunes the model to balance between denoising the observed constrained structures and fully reconstructing them.

## 3.2 IO-LVM in path inference

Consider a direct graph containing the set of edges $E$ and the set of nodes $V$. Let $\mathcal{X}$ be a set of possible paths $\boldsymbol{x}$ in this graph. More precisely, $\mathcal{X} = \mathcal{X}' \cup \mathcal{S}$, where $\mathcal{X}' \subseteq \{0, 1\}^{|E|}$ is a set of binary vectors, where each $\boldsymbol{x}' \in \mathcal{X}'$ is an $|E|$-dimensional vector representing the usage of edges in a path (1 for used edges and 0 otherwise); and $\mathcal{S} := \{(s, t) \mid s, t \in V, s \neq t\}$, defining the start and target nodes of a path. Also, Let $\omega$ be a black-box shortest path solver, e.g., that takes $s$, $t$, and edges cost $\boldsymbol{y} \in \mathbb{R}_{>0}^{E}$ as inputs[1], so that $\hat{\boldsymbol{x}} := \omega(\boldsymbol{y})$. More specifically, the COP for the shortest path problem can be formulated with a linear objective: $\hat{\boldsymbol{x}} \in \operatorname{argmin}_{\boldsymbol{x} \in \mathcal{X}} \langle \boldsymbol{y}, \boldsymbol{x} \rangle$. As done in Berthet et al. (2020), for a smooth mapping and in order to leverage the linear gradient of the Fenchel-Young loss as described in Subsection 2.2, a perturbed argmin is defined as $\hat{\boldsymbol{x}}_\epsilon := \mathbb{E}_\epsilon [\operatorname{argmin}_{\boldsymbol{x} \in \mathcal{X}} \langle \boldsymbol{y} + \epsilon, \boldsymbol{x} \rangle]$. Taking into account the method description, and considering that $\boldsymbol{y}^\theta$ is sampled from $p_\theta(\boldsymbol{y} \mid \boldsymbol{z})$, Equation 3 is rewritten as

$$l(\theta, \phi) = \mathbb{E}_{q_\phi(\boldsymbol{z}|\boldsymbol{x})} \left[ \langle \boldsymbol{y}^\theta, \boldsymbol{x} \rangle - \langle \boldsymbol{y}^\theta, \hat{\boldsymbol{x}}_\epsilon^\theta \rangle \right] + \beta D_{\text{KL}} \left( q_\phi(\boldsymbol{z} \mid \boldsymbol{x}) \| P(\boldsymbol{z}) \right). \tag{4}$$

Here, $\boldsymbol{y}^\theta$ is interpreted as the inferred edges (transitions) costs in the graph, while $q_\phi(\boldsymbol{z} \mid \boldsymbol{x})$ is interpreted as the encoded information of these costs.

Algorithm 1 details the steps in the training process in a stochastic gradient descent (SGD) fashion using an encoder $h_\phi$ to model $q_\phi(\boldsymbol{z} \mid \boldsymbol{x})$ and a decoder $g_\theta$ to model $p_\theta(\boldsymbol{y} \mid \boldsymbol{z})$. In the algorithm, step 4 decodes from the latent space and makes sure that transition costs are non-negative as input to a path solver, e.g., Dijkstra. In Step 5, $\hat{\boldsymbol{x}}_\epsilon$ can be computed by sampling a single noisy solution instead of estimating $\mathbb{E}_\epsilon$. This keeps the process simple yet effective in the long term due to the use of SGD. Note that Step 7 contains the backpropagation of the analytical gradient of the reconstruction loss w.r.t. elements in $\boldsymbol{y}$ as presented in Subsection 2.2, i.e., $\nabla l_{\text{FY}}^\epsilon(\boldsymbol{y}, \boldsymbol{x}) = \boldsymbol{x} - \hat{\boldsymbol{x}}_\epsilon$. Once the algorithm is trained, we can reconstruct paths from parts of the low-dimensional latent space using $g_\theta$, e.g., sampling from different parts of the latent space, so that we can see the different patterns reconstructed in the path space.

## 4 Experiments

The experiments focus on path planning in graphs using Dijkstra's algorithm for the shortest path problem. Two datasets, described in Subsection 4.1, are used under the assumption that agents optimize paths based on their transition costs. Subsection 4.2 analyzes the interpretability of the learned latent vectors; Subsection 4.3 demonstrates the latent space's ability to reconstruct accurate paths; Subsection 4.4 examines the impact of $\beta$ on latent space projection and reconstruction; and Subsection 4.5 evaluates overall performance, comparing it to conceptual baselines.

### 4.1 Datasets

In the following paragraphs we briefly explain the used graphs and datasets. Further details of each dataset generation or preprocessing are provided in the code in the supplementary material.

---

[1]The "unconstrained" space $\mathcal{Y}$ is actually dependent on the input space of the COP solver. As in our purposes we are dealing with non-cycling paths and Dijkstra, we assume (and ensure) that the values in this space is always greater than zero.

---

**Algorithm 1** One epoch of the training process: IO-LVM for path inference

---

1: **Components:**
2:     - Encoder $h_\phi$; Decoder $g_\theta$.
3: **Input:** Dataset $\mathcal{D} = \{(\mathbf{x}'_i, s_i, t_i)\}_{i=1}^N$
4: **Output:** Trained model parameters
5: **for** each sample $(\mathbf{x}', s, t) \in \mathcal{D}$ **do**
6:     **Step 1:** Form the path information vector $\boldsymbol{x} = \text{concat}(\boldsymbol{x}', s, t)$.
7:     **Step 2:** Encode $\boldsymbol{x}$ using $h_\phi$ to obtain the latent mean and variance: $(\mu, \sigma) = h_\phi(\mathbf{x})$.
8:     **Step 3:** Sample $\boldsymbol{z}$ using the reparameterization trick: $\boldsymbol{z} = \mu + \sigma \cdot \epsilon$, where $\epsilon \sim \mathcal{N}$.
9:     **Step 4:** Map $\boldsymbol{z}$ to the edges cost space using $g_\theta$: $\boldsymbol{y}^\theta = \max\{g_\theta(\boldsymbol{z}), 0\}$.
10:     **Step 5:** Solve shortest path using $\omega$ for the inferred path: $\hat{\boldsymbol{x}}_\epsilon^\theta = \omega(\boldsymbol{y} + \epsilon)$, where $\epsilon \sim \mathcal{N}$.
11:     **Step 6:** Compute the loss for this sample as described in Equation 4.
12:     **Step 7:** Update model parameters using the computed loss.
13: **end for**

---

It is noteworthy that all graphs are direct. We hide the edges direction in the figures for a better visualization.

**Synthetic Waxman Random Graph**  We generate a Waxman graph (Van Mieghem, 2001) with 700 nodes ($\alpha = 0.05$, $\beta = 0.6$), where the probability of an edge between two nodes $u$ and $v$ is given by $P(u, v) = \alpha \cdot \exp\left(-\frac{d(u,v)}{\beta \cdot d_{\max}}\right)$, where we considered $d(u, v)$ as the Euclidean distance between nodes $u$ and $v$, and $d_{\max}$ is the maximum distance between of two nodes, consequently ending up in 7230 edges. We create three edge cost sets to simulate three different agents performing decisions to go from start and end nodes. For each agent, we add a random noise in the cost so the generated paths are different from each other. The edge costs are based on Euclidean distances, with higher costs for the southern edges for agent 1, and higher costs in the northern for agent 3. Two sets of 6,000 observed paths are generated: one with a single source and target pair (Figure 2a, left) and another with multiple source-target pairs (Figure 2b, right). Further details on cost generation are provided in the code.

**Ships dataset**  We use the Automatic Identification System (AIS) data provided by the Danish Maritime Authority (Danish Maritime Authority, 2020), considering latitude and longitude projected in a 2D space for simplicity. The analysis focuses on paths from the first week of the months January 2024, May 2024, and June 2024. Only paths that exceed a distance of 4 units (in latitude/longitude) in Euclidean space are included. A path is considered completed either when the ship speed approaches zero or when there is an abrupt change in its heading. In some cases, there are gaps in the latitude/longitude signals; when such jumps occur, we segment the data and treat them as separate paths. We created a grid graph with a distance of 0.09 units between adjacent nodes, focusing on the area where there are more route options to be taken, which in total led to 2513 nodes and 8924 edges.

## 4.2  ENCODING PATHS TO LATENT SPACE

**Synthetic Paths Encoding.**  We used the learned $h_\phi$ to map the paths in the test dataset to the latent space. Figure 2 illustrates this mapping on the synthetic dataset, where the latent space is restricted to two dimensions. The colors of each point in the latent space illustrate which agent performed the task. It is important to remind that the agent information was not provided in training, this is only for interpretation purposes. IO-LVM successfully disentangles the factors associated with the costs of three different agents. This disentanglement is evident not only when the dataset contains observed paths between a single pair of start and target nodes (Figure 2a) but more importantly when multiple pairs of start and end nodes are present (Figure 2b). The example with multiple pairs is important because it highlights that IO-LVM is capable to encode the underlying transition costs if there is enough data. As an example, there are multiple different red paths, even with different start and target nodes, but they are mapped in the same region in the latent space because they share similar underlying transition costs.

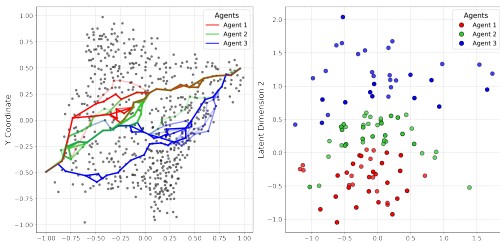 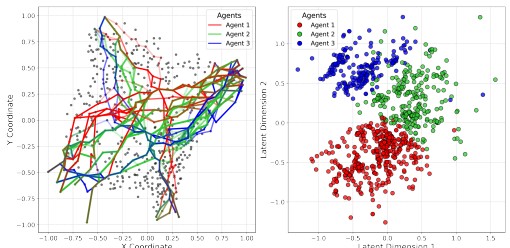

(a) Left chart is an illustration of paths in the $\mathcal{X}$ space. Right chart is the embedding of each path to the latent space $\mathcal{Z}$ using $h_\phi$.

(b) Left chart is an illustration of paths in the $\mathcal{X}$ space. Right chart is the embedding of each path to the latent space $\mathcal{Z}$ using $h_\phi$.

Figure 2: Latent space embedding of paths after training for a *single* (a) pair of start and target nodes and for multiple (b) pairs of start and target nodes. The figure is better visualized with colors.

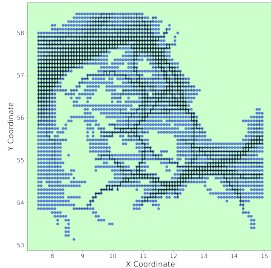 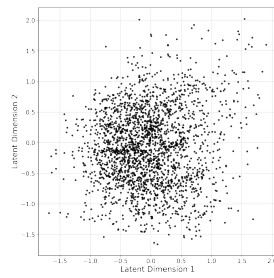 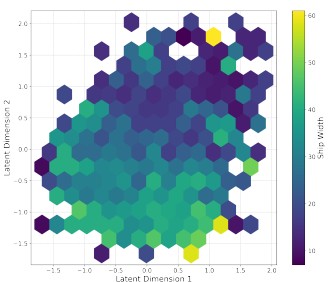

(a) Paths in the ship dataset are in black. Blue circles correspond to the graph nodes.

(b) Ship paths mapped to a 2-dimensional latent space after parameters $\theta$ and $\phi$ are learned.

(c) Plot of ship's width average in each hexagon of the latent space.

Figure 3: From (a) to (b), paths are projected to the latent space using the mean of $q_\phi$. Ship width, although not used in training, are observed in (c) as a captured feature in the latent space.

**Ship Paths Encoding.** Figures 3a and 3b show the mapping using the ship dataset for latent dimensions 1 and 2 (we observed that increasing the number of dimensions did not help for better performance, Figure 8 shows that using 3 dimensions ended up in a high correlation between dimensions 2 and 3). For this dataset, instead of different types of agents performing path decisions, we bring the information of ship width in Figure 3c. Each hexagon corresponds to a small subspace in the latent space. For each hexagon, the average of the ships' width are computed and plotted with a color map. Here, there is a subtle trend related to ship width within the latent space; larger ships are less frequently found in the top-right corner of the graph, leading to a low average ship width in that region. This is another example that IO-LVM was capable to capture unobserved factors within the latent projection.

It is important to note that the proportion of non-observed paths in the test set is high in the synthetic data involving multiple start and end node pairs, and in the ship dataset, where there is typically only a single or few observed path between distinct node pairs. This means that most of the paths mapped in Figure 2b and 3a were not observed during the training process.

## 4.3 RECONSTRUCTING FROM LATENT SPACE

**Reconstruction from parts of the latent space.** In order to illustrate reconstructions from parts of the learned latent space for the dimensions $l_1$ and $l_2$, we sample 20 times from different 2D independent Gaussian distributions with mean $(\mu_1, \mu_2)$ and identical standard deviations $(\sigma, \sigma)$. The reconstruction is performed with the learned $g_\theta$ with these samples as input, and then Dijkstra is called to output paths between desired start and end nodes. The circles in the latent space plots (top

row) in Figures 4 and 5 represent the region bounded by $\mu_1, \mu_2$, and $2\sigma$, i.e., $(l_1 - \mu_1)^2 + (l_2 - \mu_2)^2 \leq 4\sigma^2$.

**Reconstruction for Synthetic Paths.** The resulting reconstructed synthetic paths are shown in the bottom graphs of Figure 4. It can be observed that points closer in the latent space share a relatively high number of edges in the graph. Additionally, as $\sigma$ increases, the number of distinct reconstructed paths naturally grows, e.g., difference between the third and fourth columns in Figure 4. Note that the Dijsktra in the loop ensures that all reconstructed paths remain valid.

**Reconstruction for Ship Paths.** A similar process is applied to the ship dataset and can be observed in Figure 5. An interesting pattern emerges here: Some regions of the latent space containing wider ships avoid the Copenhagen canal (Oresund Strait) when traveling from the east to the north part of Denmark even though it is the shortest path in terms of euclidean distance, as observed in the second column of the figure where ships prefer going through the Great Belt. This is for example, a different decision from ship paths observed in the first column of the figure, where the preference is through the Oresund Strait.

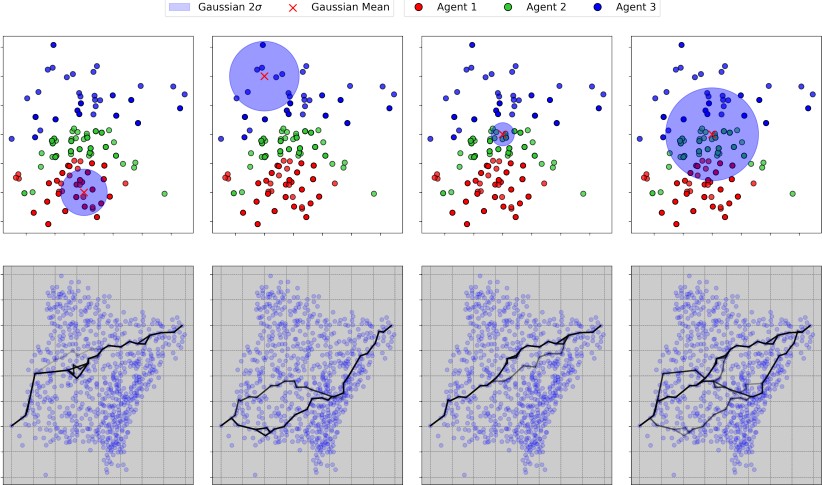

Figure 4: Reconstruction for the synthetic data with single pair of start and end nodes. Top charts: region of samples from a Gaussian in the latent space. Bottom charts: corresponding generated trajectories. Blue agents has higher costs on edges in the north, while red edges has higher costs on edges in the south. The figure is better visualized with colors.

**Unsupervised Learning facilitation** Mapping the data to a low-dimensional latent space simplifies the application of unsupervised learning techniques. One straightforward example is illustrated in Figure 9 (Appendix), where we perform a simple clustering using K-Means in the latent space. The corresponding clusters are then mapped back to the path space, demonstrating their similarity.

### 4.4 EFFECT OF VARYING $\beta$: DENOISING VERSUS RECONSTRUCTION

We analyze the effect of varying $\beta$ on three metrics in a synthetic dataset with a fixed start and target node: the number of distinct paths reconstructed by the decoder using the test dataset, the Fenchel-Young loss and the Intersection over Union (IoU) metric between observed and inferred edges usage during training (An illustration of the correlation between FY loss and IOU is shown in the Learning curve, Figure 7 in Appendix). Table 1 summarizes the impact of increasing $\beta$. As $\beta$ increases, the number of distinct paths decreases, indicating a denoising effect due to the diminished influence

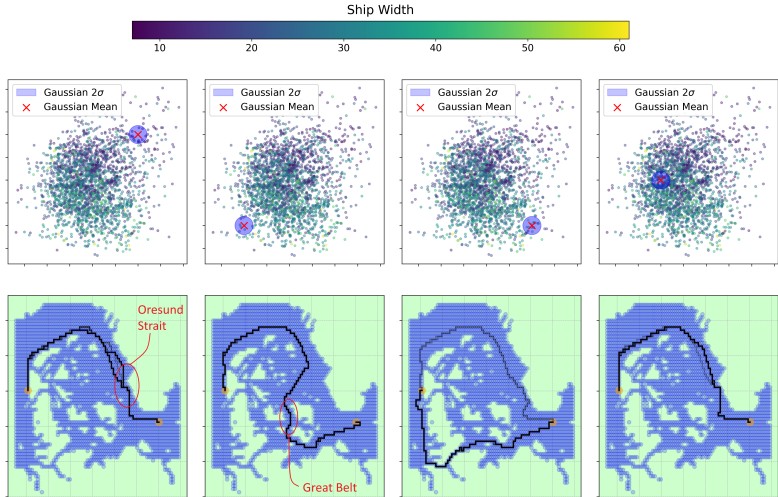

Figure 5: Reconstruction for the ship dataset. Top charts: region of samples from a Gaussian in the latent space. Bottom charts: corresponding generated trajectories in the graph given a hypothetical (non-existent in the training paths) pair of start and target nodes.

Table 1: Effect of varying $\beta$ on path reconstruction. Lower $\beta$ yields more distinct paths, while a balanced $\beta$ enables denoising. Higher $\beta$ leads to posterior collapse.

| $\beta =$ | 1e-5 | 1e-4 | 5e-4 | 1e-3 | 5e-3 | 1e-2 | 5e-2 | 1e-1 |
|---|---|---|---|---|---|---|---|---|
| Distinct paths | 66 | 59 | 51 | 38 | 15 | 13 | 4 | 1 |
| FY train loss | 0.021 | 0.022 | 0.027 | 0.032 | 0.049 | 0.054 | 0.099 | 0.150 |
| IoU train | 0.973 | 0.981 | 0.975 | 0.963 | 0.940 | 0.931 | 0.832 | 0.491 |

of the reconstruction loss. This results in the decoder reducing diversity of generated paths due to the posterior collapse. The Fenchel-Young loss increases and the IoU decreases with larger $\beta$, also reflecting a reduction in reconstruction accuracy. These trends highlight a trade-off: higher $\beta$ values favor denoising over reconstruction fidelity, while lower values focus on better reconstruction.

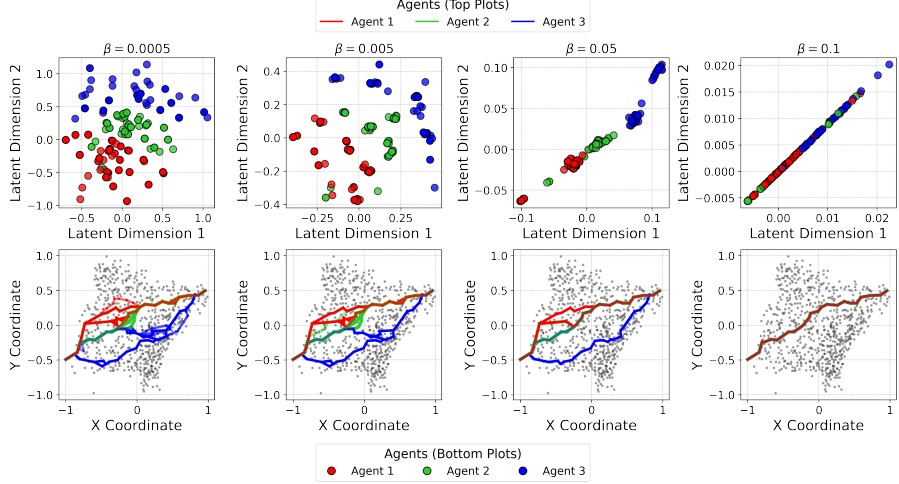

Figure 6: Varying $\beta$ in the latent space projection (top graphs) and in the reconstruction (bottom graphs).

Table 2: Results on the reconstruction of paths distribution and path prediction.

| Method | Synthetic | | Ship |
|--------|-----------|---------|------|
| | $D_{\text{JS}}$ | Spearman | $D_{\text{JS}}$ per sample |
| PO | $0.058 \pm .008$ | $0.813 \pm .025$ | $0.500 \pm .161$ |
| VAE | $0.112 \pm .002$ | $0.639 \pm .144$ | No convergence |
| IO-LVM | $0.056 \pm .003$ | $0.873 \pm .016$ | $0.467 \pm .195$ |

## 4.5 PREDICTIVE AND RECONSTRUCTION RESULTS

**Baselines** We consider two conceptual baselines. The first is based on the method from Berthet et al. (2020), which we refer to as Perturbed Optimizer (PO). We modified the original method in two ways: (1) we learn based on paths without considering context, as the original paper is context-based; and (2) to promote distribution reconstruction, we re-introduce the noise $\epsilon$ during inference, similar to its use in the training process, to account for variability in the path space. The second baseline is a traditional $\beta$-VAE without the constrained mapping, where the autoencoding occurs directly in the path space, $\mathcal{X}$.

**Metrics and Results for the Synthetic Dataset** For the synthetic data with a single start-target pair, two metrics are evaluated: the Jensen-Shannon divergence ($D_{\text{JS}}$, lower is better) between edge usage in 1,000 test samples and 1,000 reconstructed paths, indicating the similarity of edge frequencies, and Spearman's rank correlation (higher is better) between the common paths in the inferred and actual set of paths to assess the alignment in frequency ranking. Each method is sampled five times to compute the mean and standard deviation. IO-LVM outperforms PO in Spearman's correlation, due to its ability to recover distinct costs in the unconstrained space, even in multimodal cases (e.g., three agents with different paths). In contrast, PO generates noisy paths around a (single) learned optimal set of transition costs, $\omega(\boldsymbol{y} + \epsilon)$, which may not align with the true distribution. The $\beta$-VAE, despite good training performance and a well-structured latent space, failed to reconstruct valid paths, indicating poor generalization.

**Metrics and Results for the Ship Dataset** In the ship dataset, paths include multiple start and end nodes, making it infeasible to measure distribution distances for fixed start-target pairs due to the limited number (or even a single) of available paths per pair. Therefore, $D_{\text{JS}}$ is measured between the edges of each inferred sample and its corresponding test sample, and the average is computed across the dataset. For this evaluation, the most likely path from each model and baseline is compared to the observed paths. IO-LVM slightly better than PO, but the difference is not statistically significant due to high variance in the error metric. The $\beta$-VAE baseline failed to converge, likely due to the graph size and the complexity of multiple start and target node scenarios.

## 5 CONCLUSION

This paper proposed IO-LVM, a novel approach for learning latent representations of constrained optimization problem (COP) costs, specifically for path planning in graphs. The method leverages amortized inference and integrates a shortest path solver within a probabilistic framework, allowing for the modeling of multiple agents and diverse behaviors in graphs. By employing a Fenchel-Young loss with perturbed inputs, it overcomes the gradient challenges in optimizing COPs, ensuring feasible and interpretable path reconstructions. The learned latent space captured meaningful structures, highlighting the model's characteristic to distinct agent behaviors, while maintaining accurate path reconstruction and prediction. The study also explored the role of the $\beta$ hyperparameter on using the model for denoising paths or aiming full reconstruction. Comparisons with baselines further validated its performance in path distribution reconstruction and prediction. Our method description is valid for a general set of COPs if gradient estimation is available. By leveraging different types of gradient estimation, a future work could extend this framework to incorporate more complex decision-making scenarios.

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

## A    ADDITIONAL FIGURES

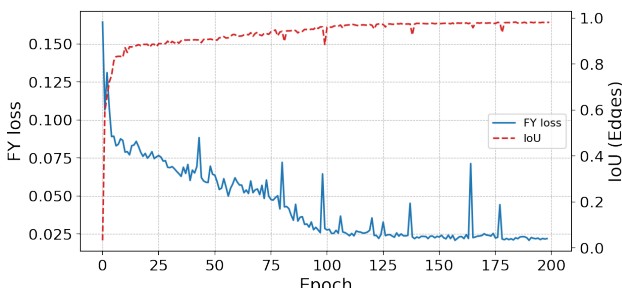

Figure 7: Fenchel-Young loss and IoU between edges computed during the training process.

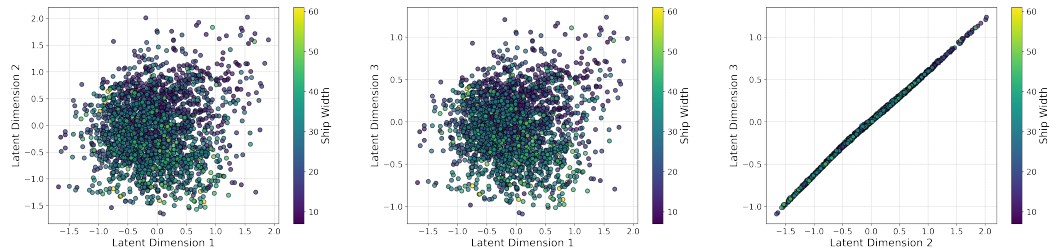

Figure 8: Latent space of ship trajectories using three dimensions. The right graph indicates that there is no need for a third latent dimension. Narrow ships are more concentrated in the top right corner of the two left graphs. The colorbar is only for an evaluation purposes, since the agent type is not given to the training process.

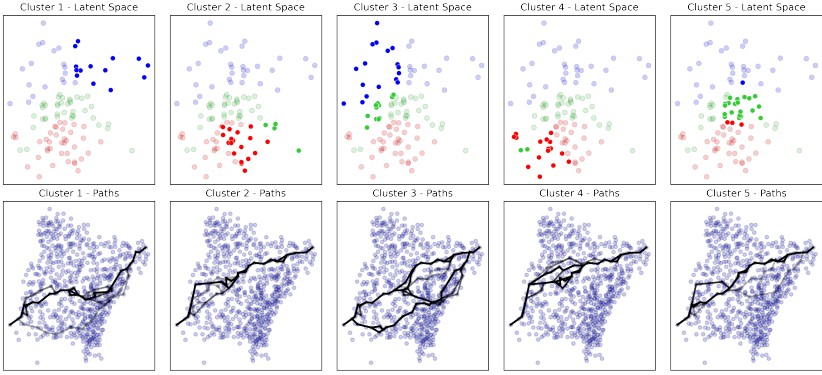

Figure 9: Clustering the latent vectors and visualizing the correspondent paths.

