# OpenReview forum: "IO-LVM: Inverse optimization latent variable models with applications to inferring and explaining paths"
_ICLR.cc/2025/Conference — Submitted to ICLR 2025_

### Official Review · Reviewer_5TkB · 2024-11-03

**Soundness:** 3
**Presentation:** 4
**Contribution:** 3
**Rating:** 5
**Confidence:** 2

**Summary:**

The paper introduces a novel approach called Inverse Optimization Latent Variable Model (IO-LVM), designed to learn interpretable latent representations of cost functions underlying constrained optimization problems (COPs) based on observed decisions, specifically in path-planning tasks. Traditional methods struggle with structured outputs or assume a single underlying cost function, limiting their ability to e.g. learn from multiple agents making decisions. IO-LVM overcomes this by using a low-dimensional latent space that captures varying cost structures, utilizing a Fenchel-Young loss and a COP solver for gradient estimation. This approach enables tasks like clustering paths, denoising, and path prediction for new start-target pairs, demonstrated on synthetic and real-world ship routing data. Overall, the model is rather interpretable and flexible, which makes it a robust tool for complex path inference and for understanding agent-specific behaviors in path planning.

**Strengths:**

- The preliminary section is clear and well-written.
- The method section is very well presented and easy to understand.
- The beta ablation (table 1) makes sense.

**Weaknesses:**

- The coefficient $\beta$ is presented as "introduced" in this work to trade off the reconstruction of the data and how Gaussian the latent distribution is. In the original VAE paper [Kingma and Welling, 2014], this coefficient is indeed 1. To my knowledge, setting this coefficient to less than 1 is not new and is rather well-known in the VAE community because the encoder learning signal from the KL is much stronger than from the reconstruction loss. Examples include the beta-VAE paper [Higgins et al., 2017] that you also mentioned, and most VAE implementations (e.g. https://github.com/AntixK/PyTorch-VAE). It seems that introducing this coefficient as something new is an inaccuracy.
- Most figures from the experiment section are hard to interpret by someone who isn't familiar with the tasks used for demonstration.
- Comparison with baselines (especially PO) is hard to make sense of and does not seem very significant (table 2).

**Questions:**

1. As mentioned above, the comparison with baselines is hard to interpret. How significant are IO-LVM results compared to PO?
2. Are latent spaces always of dimension 2 in the paper? How has experimenting with higher dimensions looked? What is the conclusion of needing such low-dimensional spaces?

---

> ### Author Response · Authors · 2024-11-14
>
> Dear Reviewer, thank you for your feedback. We hope the following responses clarify your points of concern:
>
> Q: "The coefficient β is presented as ‘introduced’ to balance data reconstruction with a Gaussian latent distribution."
>
> A: This is not our intention, and we will be glad to clarify better the writing if this is the interpretation. Our intention is to claim that we generalize the β parameter for a constrained optimization scenario.
>
> Q: "The comparison with baselines is hard to interpret. How significant are IO-LVM results compared to PO?"
>
> A: Unlike PO (and similar state-of-the-art methods), IO-LVM can model multiple cost functions. This capability is unique to our approach, this is why we bring this comparison.
>
> Q: "Are latent spaces always 2-dimensional in the paper? What conclusions can be drawn from using low-dimensional spaces?"
>
> A: The main challenge in our problem is path structuring, which is addressed by Dijkstra’s explicit constraints. Unlike traditional VAEs, which require higher-dimensional spaces to reconstruct complex structures (e.g., images), our VAE only needs to reconstruct a simpler, unconstrained edge cost space. This is why 2 / 3 dimensions are generally enough. More dimensions would not me harmful though (as we have observed), with a good choice of beta.

---

> > ### Comment · Reviewer_5TkB · 2024-11-24
> >
> > Thank you for your answers, they have clarified some of the earlier-mentioned points of concern. I have decided to maintain the original review score with the main point of concern being the experiment section where the comparison to the baseline and the different figures are hard to interpret. I believe it could be improved and become a stronger work for another venue.

---

### Official Review · Reviewer_FgFv · 2024-11-04

**Soundness:** 2
**Presentation:** 3
**Contribution:** 2
**Rating:** 3
**Confidence:** 3

**Summary:**

The paper proposes IO-LVM, a solution to constrained optimization problems (COPs) with unknown cost functions that leverages VAE and Fenchel-Young loss to learn an informative latent representation of graph paths. IO-LVM is evaluated on synthetic and real-world ship path datasets and can disentangle a dataset's factors of variations, denoising observed paths, and predicting optimal paths for unseen data.

**Strengths:**

- Training a neural generative model with COP solvers is to my knowledge novel and interesting.
- The paper shows that the trained VAE's latent embeddings capture meaningful details about the data-generating process, such as the agent index and ship width.
- The paper presents a comprehensive ablation and qualitative analysis of simple pathfinding problems.

**Weaknesses:**

- The motivation behind the paper's research direction could be stated better. The introduction elaborates on IO-LVM's interesting capabilities in detail, but none of them are practically relevant in their current state. Further elaborating on how they can lead to effective solutions for real-world problems would make the paper stronger.
- The paper leans too heavily on qualitative analysis over quantitative analysis. Only one table of results compares against baselines, and IO-LVM's performance gain is only meaningful on the synthetic dataset's Spearman metric. It is also unclear what different Jensen-Shannon divergence values tell us about a model's predictive and reconstructive abilities. Having additional quantitative metrics, preferably interpretable ones, would be beneficial.
- The proposed experiments are relatively simple. Given that IO-LVM's latent space clusters agent ID variable significantly better for the simpler synthetic paths datasets compared to the ship width variable for the more complicated Ships dataset, I question whether IO-LVM will scale well to more complicated problems.

**Questions:**

- For the Ships dataset, why do you think having just two latent dimensions was sufficient? Is this related to the fact that the problem is fairly simple?
- Do you know of any other metric aside from Jensen-Shannon divergence and Spearman's rank correlation that can better highlight IO-LVM's advantage?
- What other problems aside from path explaining do you think IO-LVM can be applied to with small modifications?

---

> ### Author Response · Authors · 2024-11-14
>
> Dear Reviewer, thank you for your constructive feedback. We hope the following responses clarify some key aspects of our work:
>
> Q: "The paper leans too heavily on qualitative analysis over quantitative analysis. Only one table of results compares against baselines, and IO-LVM’s performance gain is only meaningful on the synthetic dataset’s Spearman metric."
>
> A: One of the focus of this paper was indeed to provide more qualitative analysis on the latent space, which we believe it is not a weakness. While there are many paper providing tables and numbers, we preferred to bring a more insightful perspective in the unsupervised setting, together with the reconstruction table results.
>
> Q: "It is unclear what different Jensen-Shannon divergence values tell us about a model’s predictive and reconstructive abilities. Additional quantitative metrics would be beneficial."
>
> A: Selecting a reconstruction metric for paths is challenging. Previous works (e.g., https://arxiv.org/abs/2002.08676 and https://arxiv.org/abs/1912.02175) often use “full path match” metrics (1 if the path fully match, 0 otherwise, and then average), but these are unreliable for larger graphs with noisy paths, where full path matching would be near 0% despite of the method chosen. Instead, we chose JS divergence for its symmetry and ability to quantify probability distribution distance between inferred and actual edges/paths choice. One could think of Euclidean metrics also, but they would not make much sense in our case because we wanted to show a "discrete" quantification, e.g., closer nodes in distance are as wrong as distant nodes. We are open for suggestions in this regard.
>
> Q: "The proposed experiments are relatively simple. Given that IO-LVM’s latent space clusters agent ID significantly better on simpler synthetic datasets than on the Ships dataset, I question its scalability."
>
> A:  As the reviewer mentioned, the ship dataset is a complicated dataset. We believe the ship dataset is complicated enough to test the scalability of IO-LVM. The reason why Figure 3c (showing the separability in the latent space in the ship data) is not as clean as Figure 2b (showing the separability for the agents) is because there might be many other variables impacting the path choice in the ship dataset, even variables (or noise) that we do not have access to for a possible evaluation.
>
> Q: "For the Ships dataset, why do you think having just two latent dimensions was sufficient? Is this related to the problem’s simplicity?"
>
> A: The main challenge in our problem is path structuring, which is addressed by Dijkstra’s explicit constraints (or potentially any other solver for different problems). Unlike traditional VAEs, which require higher-dimensional spaces to reconstruct complex structures (e.g., images), our VAE only needs to reconstruct a simpler, unconstrained edge cost space.
>
> Q: "Do you know of any other metric aside from Jensen-Shannon divergence and Spearman’s rank correlation that can better highlight IO-LVM’s advantage?"
>
> A: This is addressed in the response above.
>
> Q: "What other problems aside from path explanation could IO-LVM be applied to with small modifications?"
>
> A: With slight modifications, IO-LVM could apply to any problem where the COP can be formulated as an Integer Linear Program with an efficient solver available.

---

> > ### Comment · Reviewer_FgFv · 2024-11-23
> >
> > Thank you for your detailed response. I agree that the paper's detailed qualitative analysis is a strength; however, I find the paper's limited quantitative analysis to be a significant weakness. Reporting additional metrics such as graph edit distance [1] (or other metrics employed by the graph machine learning community) would better convey IO-LVM's advantage over VAE without penalizing minor deviations in the reconstructed paths. As per why it was sufficient to have just two latent dimensions for the Ships datasets, I still believe that the problem's relative simplicity is the primary reason. For truly complex path inference problems (ex. same ship routing problem for the entire Pacific Ocean), I can't imagine that two latent dimensions are enough for path reconstruction. Given a reasonable alternative explanation, ideally with evidence, I am open to changing my mind on this topic. However, given that my reasoning holds, I believe the proposed experiments are too simple to effectively highlight IO-LVM's potential in solving real-world problems.
> >
> > [1] Blumenthal, D. B. (2019). New techniques for graph edit distance computation. arXiv preprint arXiv:1908.00265.

---

### Official Review · Reviewer_z9GS · 2024-11-04

**Soundness:** 3
**Presentation:** 3
**Contribution:** 2
**Rating:** 5
**Confidence:** 3

**Summary:**

I found the rebuttal to be convincing, so I raised my score.
---

-The paper’s central motivation is that VAEs (or generative models in general) will generate infeasible paths.  I don’t think this is really validated experimentally in the paper.
  -While the proposed method is neat, it’s not clear to me that it justifies the substantial added complexity.


notes from reading paper:
  -Learning representations from solutions of constrained optimization problems (COPs) with unknown cost functions is challenging.  A VAE may fail to capture the constraints.
  -Paper proposes inverse optimization latent variable model.  Latent space model of COP costs, which are then reconstructed via a COP solver.  Leverage gradients of a Fenchel-Young loss through deterministic solver.
  -Synthetic and actual ship routing data.  Validate on Dijkstra path planning problems.
  -Fenchel-Young loss is defined as gap between the score function and the score function under a regularized solution based on the score vector.  Using a linear cost function, we have a loss which is minimized only if the regularized solution and the given solution are the same.
  -Map x to latent space z, then to unconstrained space y, then to constrained space x.

**Strengths:**

The problem of learning generative models of optimal paths is interesting.

**Weaknesses:**

The method seems unappealing because of the complexity of needing to run a pre-defined planning algorithm (such as Dijkstra's).  Additionally, the paper doesn't carefully validate the claim that VAEs and other generative models produce paths which violate constraints.  I'm not fully convinced of this.

**Questions:**

-Is it possible to use the algorithm to learn optimal trajectories even when trained on sub-optimal data?  I think this aspect could be interesting.

---

> ### Author Response · Authors · 2024-11-14
>
> Dear Reviewer, we value your detailed feedback. We are open for improving our work, however, we think that there were some misunderstandings regarding the content of our proposed method and the topic in general, so we would like to clarify it better.
>
> Q: "The method seems unappealing because of the complexity of needing to run a pre-defined planning algorithm (such as Dijkstra's)."
>
> A: Many studies in this field incorporate these algorithms within neural network frameworks, i.e., they also have to run does algorithms during training and inference. We could list at least 50 papers in this area, but here are some of the important works:
> https://arxiv.org/abs/2002.08676
> https://arxiv.org/abs/1710.08005
> https://arxiv.org/abs/1912.02175
> We hope you can reconsider whether this aspect is truly a weakness in the paper.
>
> Q: "The paper doesn’t carefully validate the claim that VAEs and other generative models produce paths which violate constraints. I’m not fully convinced of this."
>
> A: It is well known that neural networks cannot ensure COP feasibility. VAEs do not impose explicit constraints; instead, they would need substantial data to infer constraints, with no guarantee of generalization. By integrating a Dijkstra (or other structured algorithm) block, we ensure that the constraints are explicitly modeled, ensuring feasibility. This challenge is reflected in Table 2, where VAE results are significantly less effective due to their lack of path structure representation.
>
> Q: "Is it possible to use the algorithm to learn optimal trajectories even when trained on suboptimal data? I think this aspect could be interesting."
>
> A: Yes, and we explored this in our work. The ship paths in our dataset are not always optimal and include noise and therefore suboptimal characteristics. Similarly, in the synthetic data, we introduce three agents with different cost functions and inject noise into each function to generate suboptimal paths. Our model can either overfit to this noise or denoise (through variations in the beta parameter).

---

### Official Review · Reviewer_vpb2 · 2024-11-04

**Soundness:** 3
**Presentation:** 2
**Contribution:** 2
**Rating:** 5
**Confidence:** 2

**Summary:**

This work proposes a method titled inverse optimization latent variable model (IO-LVM). This latent-variable model can model the space of solutions of constrained optimization problems. The authors test the idea on ship routing data. Here, the model learns the latent space representing transition costs of different agents in the data. Then, COP solver provides paths based on the sampled transition costs.

**Strengths:**

- The results show that the learned latent space captures the space of paths well. IO-LVM captures nuances such as big ships not passing through Oresund Straight even when it's the shortest path purely based on the data.

**Weaknesses:**

- This is not my area of expertise, so I may not be the best judge of it, but the paper was hard for me to understand.
- The experiments are done on problems with few dimensions. If I understand correctly, the latent learned latent space has only 2 or 3 dimensions. The method would be more convincing if there were experiments with more dimensions, e.g. discrete decision problems.
- The experiments are specific to paths generation. This objective is quite general, but in the experiments, the mapping from Y to X is essentially the shortest path finding algorithm.

###### writing
230 and 286: direct graph: should it be "directed" graph?

**Questions:**

1. Why do you think 2 or 3 dimensions is enough to capture the dimensionality of underlying cost functions? Is that because only 2 or 3 factors come into play in your examples, like 3 different agents in the synthetic paths setting, and size/weight of the ship in the real trajectories dataset?
2. Quantifying how well the model captures the ship width would be interesting. I'm curious whether you can learn a mapping from $z$ to the weight of the ship if that data is available.
3. What could be other applications of this method apart from paths sampling? Could this be applied to discrete decision problems from [Perturbed Optimizer paper](https://arxiv.org/pdf/2002.08676)?

---

> ### Author Response · Authors · 2024-11-14
>
> Dear Reviewer, we appreciate the time and effort you put into reviewing our paper. We hope the following clarifications will address your concerns:
>
> Q: "If I understand correctly, the latent learned space has only 2 or 3 dimensions. The method would be more convincing if there were experiments with more dimensions, e.g., discrete decision problems."
>
> A: The path reconstruction problem is inherently discrete, as the decision variables correspond to binary vectors in the edges’ space (i.e., selecting or not selecting each edge for the path). Our latent space (z) generates a continuous edge cost space. Because reconstructing edge costs is simpler than reconstructing paths, a low-dimensional latent space (2 or 3 dimensions) is sufficient. The complexity of path structure is managed by the shortest path algorithm (Dijkstra in this case), allowing the neural blocks to focus on the simpler task of decoding/reconstructing edge costs.
>
> Q: "Quantifying how well the model captures the ship width would be interesting. I'm curious whether you can learn a mapping from z to the weight of the ship if that data is available."
>
> A: We are not sure if you meant "width" or "weight," but one of our focuses here is on unsupervised learning insights. We demonstrate that latent space vectors correlate with variables (e.g., ship width) that were not part of the training data, providing evidence that the learned latent space is meaningful without explicit supervision. Having contextual features as part of the training data is not the scope of this work. Accounting them transfers the variability of the observed paths in a context-conditioned model, which is the same as done in the PO baseline.
>
> Q: "What could be other applications of this method apart from path sampling? Could this be applied to discrete decision problems from the Perturbed Optimizer paper?"
>
> A: This is a great question. Yes, this could be applied to other discrete problems that are in the Perturbed Optimizer paper. We thought that giving an emphasis in path problems is more interesting and enough to show the capabilities of our method. It is not only interesting in a graphical/illustrative perspective, but also in a real-world application for path prediction/reconstruction/denoising.

---

> > ### Comment · Reviewer_vpb2 · 2024-11-25
> >
> > Thank you for your response!
> >
> > > Q: "If I understand correctly, the latent learned space has only 2 or 3 dimensions. The method would be more convincing if there were experiments with more dimensions, e.g., discrete decision problems."
> >
> > What I had in mind here is rather the fact that in your data only few factors affect the routes. You generate 3 agents in the synthetic dataset, and for ship data, allegedly the factors that affect the path are the width of the ship and the size. My intuition is that if the behavior is more complex, it would require latent variables with more dimensions. For example, modeling car traffic in a big city would involve modeling weather, rush hours, events happening in the city, etc. If the number of these factors is big enough, then, if I understand correctly, the latent variable will need more dimensions. Is this understanding correct?
> >
> > > Q: "Quantifying how well the model captures the ship width would be interesting. I'm curious whether you can learn a mapping from z to the weight of the ship if that data is available."
> >
> > This is related to the previous question. I'm not asking to use the ship weight and width as inputs to the model, I'm just wondering if the model somehow learns these factors without them being explicitly specified anywhere in the training process. You actually address this partially in figure 3c, and show that the representation does capture the ship width to some extent.
> >
> > I maintain my score unchanged at this time. I believe this work is interesting, but the presentation and experiments could be improved.

---

### Meta-Review · Area_Chair_AVBh · 2024-12-21

**Metareview:**

This work proposes a method titled inverse optimization latent variable model (IO-LVM). This latent-variable model can model the space of solutions of constrained optimization problems. The authors test the idea on ship routing data. Here, the model learns the latent space representing transition costs of different agents in the data. Then, COP solver provides paths based on the sampled transition costs.

Reviewers agreed the setting was interesting but felt the paper could be improved in terms of both presentation and experiments.

**Additional Comments On Reviewer Discussion:**

The authors and reviewers had a discussion but reviewers remained concerned by the lack of empirical evaluation and believe the paper would be improved by more experimentation and presentation

---

### Decision · Program_Chairs · 2025-01-22

Reject